# Shaping the Inclusivity in the New Society by Enhancing the Digitainability of Sustainable Development Goals with Education

Lavinia Dovleac, Ioana Bianca Chițu, Eliza Nichifor * and Gabriel Brătucu

Faculty of Economic Sciences and Business Administration, Transilvania University of Brașov, 500036 Brasov, Romania
* Correspondence: eliza.nichifor@unitbv.ro

**Abstract:** The literature introduced the concepts of sustainability and digitalisation as one, mixed-combined and named as digitainability. By linking digital divide, sustainability, and digitalisation, the authors aimed to: (i) identify variables that might influence the digital divide index; (ii) determine variables to model sustainable development goals; and (iii) create a model to explain digitainability through education. Analysing the sample of 13 EU developed countries, the authors created three econometric models and analysed the manner of how education might shape the inclusivity in a new and ever-growing Society. The results generated an inter-connected model that shows that new Society depends on the users' medium or high skills because of their strong and positive influence on the digital divide index. The scientific contribution reveals solutions for an inclusive future, not far from the demanding requirements of the present. The theoretical implications shape the necessity to collaborate with policy makers to optimise the educational public system requirements in order to form prepared specialists for high-demanding markets in which they will work. The orientation of managerial implications of top leaders from companies must address an adapted mindset to collaborate with the academic environment to discover and understand the truth about the challenges of the new Era.

**Keywords:** digitainability; sustainable development goals; digital divide index; digital skills; society 5.0; inclusive future

## 1. Introduction

The evolution of technology cannot be stopped or avoided. It has now an organic, natural spread among humans and the development of the society itself is strongly connected to technology and digitalisation. A very important challenge for the humanity is to try to be on the same page when it comes about at the industry level. Industry 4.0 introduced the usage of smart connected technologies to improve the production manufacturing time, while industry 5.0 brings more personalization and synergy between human and machine labour with IoT technologies. Going further on the evolution path, industry 6.0 is expected to be one of renewable energy, total machine independence, interplanetary resource gathering and manufacturing [1].

Most European Union Member States are at various stages of Industry 4.0 development, while the European Commission is barely proposing strategies for Industry 5.0. In this context, Finland has a very courageous goal of becoming a strategic leader and driver towards defining Industry 6.0 [2]. Finnish industry is affected by risks caused by the pandemic, global supply chains and dependency of suppliers all around the world, and the solution to this problem could be a new industrial revolution.

Many other countries are now prepared only for the transition towards Industry 5.0, helped by on-going projects inside Horizon 2020 which contribute to the development of this concept [3]. Industry 5.0 reflects a shift from the focus on economic value to a focus

on societal value, and a shift in focusing from welfare to wellbeing [4]. More than that, Industry 5.0 moves the focus from shareholder to stakeholder value, with benefits for all concerned parties. It aims to capture the value of new technologies, providing prosperity beyond jobs and growth, while respecting planetary boundaries and placing the wellbeing of the industry worker at the centre of the production process [3].

The concepts of Society 5.0 and Industry 5.0 both refer to a fundamental shift of our society and economy towards a new paradigm. The concept of Society 5.0 was presented by Keidanren, Japan's most important business federation, in 2016. The full national transformational strategy aims to build a society that creates new values for diverse people through the full use of digital data. Digital technologies must support "human" activity and truly enrich "human" life [5]. The technology could make different domains of society more inclusive, there are studies/evidence on this aspect regarding, for instance, education [6,7], culture [7], sport [8] or finance [9].

The materials and methods used followed the steps presented in Figure 1. From the research gap identified in the literature (the digitainability applied for sustainable development goal), the literature review was performed. With valuable insights and concepts from the scientific perspective, the authors achieved the model specification by creating three regression equations transposed in three econometric models, log and linear. Reflecting the outcomes of the analysis, the authors highlighted in the next sections the digitainability concept applied to sustainable development goals through the education.

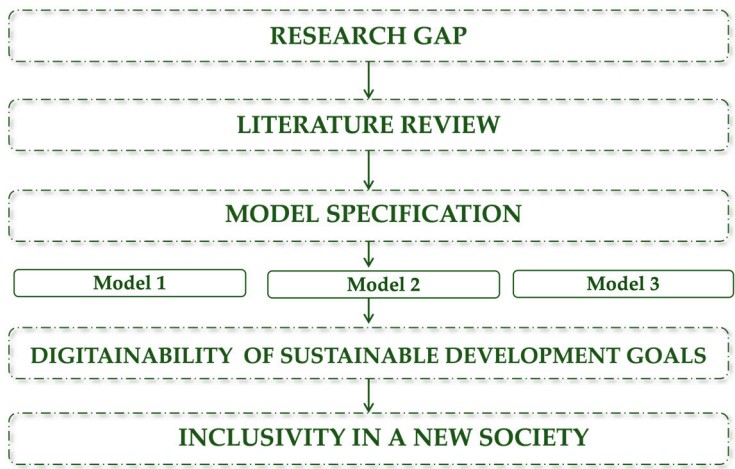

**Figure 1.** The flow of the methods used for research.

In this context, the study aims to understand how sustainability and digitalisation can create more inclusivity in the Society, by resuming it as digitainability phenomenon. For this purpose, the authors have proposed three research questions aiming to determine the factors that influence the digital divide index (hereinafter DDIX) and sustainable development goals index (SDGIX) to inter-connect the variables and find new opportunities for the future. Based on the research questions, the authors stated three main objectives. The first one (O1) was to identify variables that might influence the digital divide index, from which another one was generated. Through this, they intended to specify a model to explain the digital divide index. The second objective (O2) assumed to determine variables to model sustainable development goals and determine significant variables to explain the score of sustainable development index. The last one (O3) had the aim of inter-connecting all models with all relevant and significant variables to explain digitainability with education.

The entire framework is presented in Figure 2, where the researchers presented the objectives according to the questions and sticked them together with the results of the study for a better understanding at the first sight.

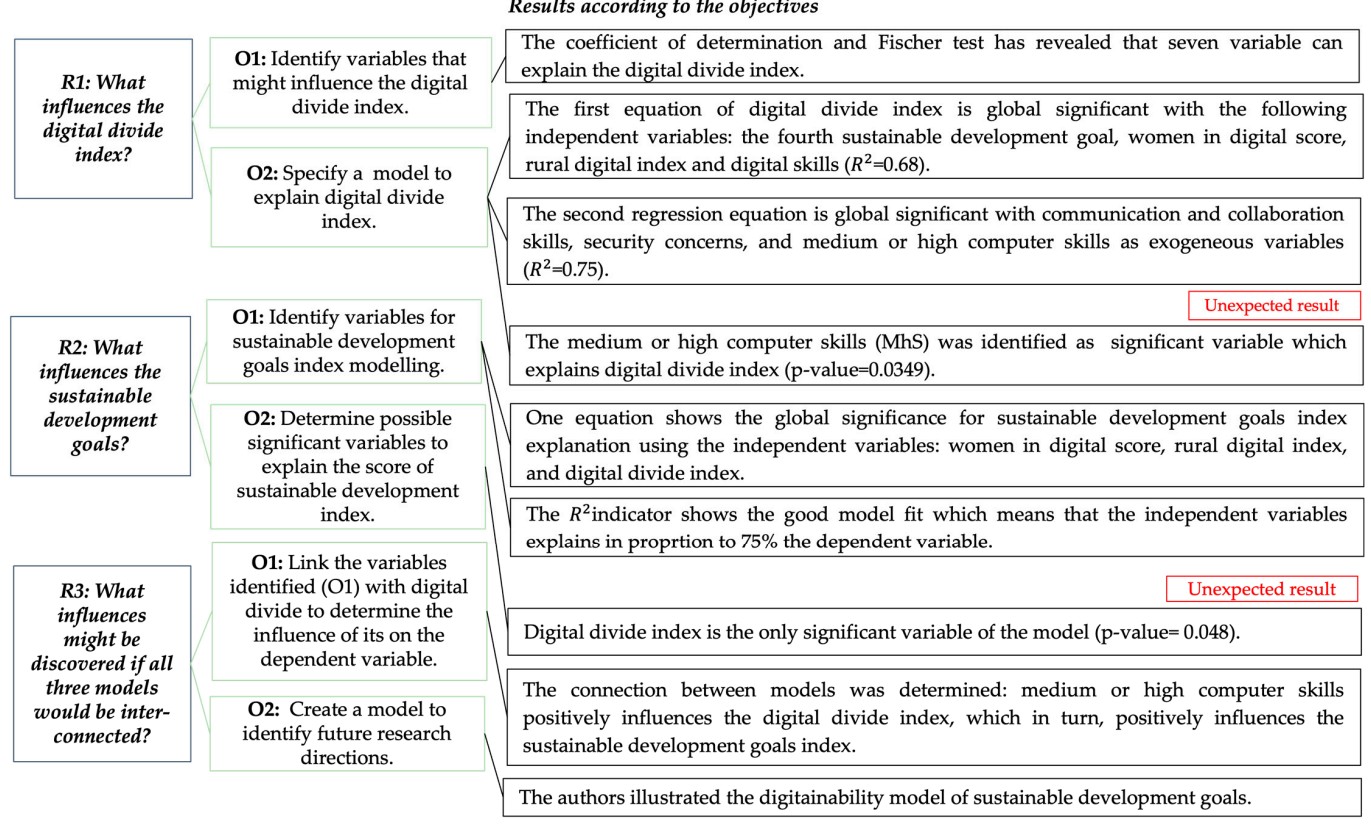

**Figure 2.** The framework of the study: research questions, objectives, and results.

The paper is structured in five parts. After the introduction of the context and the background of the issue addressed, the literature review is presented. It is followed by the materials and the methods used to perform the analysis. The findings are displayed in the dedicated section of results, wherefrom the discussions were formulated. In the final part, the conclusions of the study are underlined from the perspectives of the academic and managerial implications, limitations, and future research.

## 2. Literature Review

This Section presents the scientific progress and the perspective of Industry 5.0, as part of Society 5.0. In this regard, two sub-sections were created to summarise the relevance of the study on the topic of sustainability and digitalisation. The first one presents the Industry 5.0 and digitalisation and the second one the digital sustainable education.

### 2.1. Industry 5.0 and Digitalisation

To achieve the goals of sustainable development, Industry 5.0 brings into attention the sustainability aspect by integrating human values with technology [10]. This concept complements the concept of Industry 4.0 with worker orientation, whose decisive role was highlighted by the COVID-19 pandemic that prompted reflections to create an industrial system which is more resilient to potential shocks and to truly integrate social and environmental principles [11].

The pandemic showed that digitisation helped sectors of the economy to get through this period more easily, but at the same time, it highlighted a number of problems related to people's adaptation to the digitalisation process and related to industry resilience [12]. Moreover, the digitalisation of societies, accelerated by the COVID 19 pandemic, is an unstoppable process [13]. Industry 5.0 appeared from the necessity of bringing to the fore the needs, values of society and responsibility towards it, but these targets can only be achieved through the technological solutions offered by Industry 4.0 [14].

Studies show that there is a shift from the idea that technology is crucial to "sustainable, human-centric and resilient European industry" [3]. Thus, Industry 5.0 addresses socio-environmental concerns in the current context of industry digitisation [15], through the coexistence of three environments: the technological environment (innovation for industry development); the social environment; and the ecological environment (responsibility) [16]. The main feature of Industry 5.0 is the fact that it does not focus on the creation of goods and services for profit, but on the collaboration of companies in the industry for sustainable value co-creation [17], on the interaction of people with robots (cobots) applying innovative technologies [18].

At the same time, by integrating Industry 5.0 principles higher education institutions will be able to fully benefit from digital transformation, and by focusing on "human-oriented innovation" and by developing new cooperation models, new "channels of distribution of education, research" will be able to achieve the objectives of sustainable development [19].

Digital transformation represents a holistic and radical process of societal change, also called socio-technical transformation, which also generates certain ethical issues. The pandemic brought to attention again the fact that from a digital perspective, big inequalities still exist after more than 20 years, related to five dimensions: technical means (hardware, software), autonomy deployment, social support networks, patterns and skills [20].

### 2.2. Digital Sustainable Education in Industry 5.0

Because COVID-19 was the trigger for a more digital society, the focus from now on is on how to build a sustainable digital society wielding the so-called digitainability (digitalisation + sustainability) [13,21].

Digitisation and digital transformation are essential elements of the 2030 Agenda. They affect the university environment because education is shaped by technological developments [22]. Higher education institutions contribute to the achievement of Sustainable Development Goal no.4 (SDG4—quality education) [23]. Therefore, for a sustainable digital teaching process, teachers and students must possess a certain level of digital skills [24].

SDG 4 is particularly important for both sustainable development and the 2030 Agenda because education influences all other SDGs except SDG 14 (life under water) [25,26]. SDG 4 is statistically correlated with other SDGs, especially with SDG 1 (No poverty) and SDG 3 (good health and well-being) [27].

The pandemic accelerated the digital transformation process that had begun to take place in universities, but also brought new challenges. Face-to-face courses were replaced by those conducted in the digital space, and their format had to be reconfigured, sometimes entirely [28]. Digitisation of the learning process brings many advantages to users, but this process is still developing very slowly. Specialists believe that it should be accelerated because digital courses are seen as an opportunity for students to use new digital tools that help them organise and work independently, flexibly, and responsibly [29].

E-learning helps to fulfil SDG4 as well as to ensure the sustainability of education through the benefits it offers [30]: flexibility, inclusion, long-life learning, equality, and access to higher education.

E-learning in higher education also contributes to economic and social development by preparing a workforce with appropriate skills and preparation to cope [31].

Including the new technologies (such as augmented reality) in the teaching–learning process offers students a completely new experience of exploring the world, accessing and assimilating information. The educational experience based on augmented reality is very different from the traditional one, due to the following fundamental characteristics: collaboration capacity, continuous interaction, tangibility [32]. As an educational technology, properly applied in connection with the specific methodology, it can facilitate the teaching–learning process and make the activities more interactive and interesting. This

type of technology can also remove barriers related to special educational needs, lead to improved skills and facilitate access to information.

Teachers need quality training in the use of technology in the teaching process that covers data analysis, collaboration with the educational community implying digital technologies, content development, assessment methods, and data protection [33,34]. The existence of information and communication technologies has determined the transition from a culture centred on teaching to a culture centred on learning because it offers the possibility of adapting the teaching process to the student's learning style [34].

In the area of education, the development of knowledge production process is essential, and once again, available data represent the best asset for achieving quality education. SDG4, like all the other SDGs, can be achieved only in the context of a responsible application of technology. Digitalisation is both life-changing and disruptive. One of the tools proposed for measuring this fact and tested on SDG4 is the Digitalisation–Sustainability Matrix (DSM) [21]. At the same time, it helps identifying the SDG-indicator-specific relevance of Digitalisation and Artificial Intelligence (D&AI).

Considering the characteristics of the teaching–learning process and generations involved, a generational gap will always be present in this situation. The specialists state that the traditional age-based classification of generations is not relevant anymore for research, and it should be replaced with the following categories: digital natives and digital immigrants [35]. Digital immigrants refer to individuals born before the existence of digital technology. Due to this fact, the focus in education moves from building up stocks of knowledge (learning-about) to enabling students to participate in flows of action, where the focus is on learning to be through enculturation and on collateral learning.

## 3. Materials and Methods

Through its methods deployed, the study aims to scientifically illustrate the digitainability concept by modelling the digital divide index and the sustainable development goal no. 4 related to education [23]. In this regard, three econometric models were approached with global significance to identify the big picture of factors that can contribute for a better, sustainable, and inclusive environment in the future. For a good presentation of the methods and materials used, this section is split in two parts. The first one presents the digital divide index relevance for the study and two models with this indicator as dependent variables. The second one presents the influence of the digital divide index and other factors on education sustainable development goal.

Data were collected from secondary sources and processed for econometric analysis. In the first place, the theoretical background was consulted for correct specification of the models. After this step, datasets about relevant factors were identified and collected in separate files after the cleaning and organization process. For each indicator that belongs to the specified models was created a .csv format dataset used to run the regression analysis in EViews software. The raw data for 13 developed countries were processed and imported in the software. Each variable was transformed from natural number to natural log to change the functional form of the equation and was estimated. The results were centralised and interpreted according to the global significance (Fischer test), the significance of the independent variables (by comparing $p$-values with the level of significance). In the next sub-sections, the modelling process is described.

### 3.1. Modelling the Digital Divide Index

Digital divide has become more than a sociodemographic difference reference concept [36,37]. It has been criticised from several points of view [38] and continues to make waves in the academic, political, and socio-economic environments [39]. Described as the no. 1 threat of this century [40], it is deployed in some significant macroeconomic studies [41]. If the traditional version of the digital divide index is calculated with the percent of population that has access to fixed broadband, the speed for download and upload to fixed broadband and the number of connections to fixed broadband in residential mediums

per 1000 households [40], the 2021 DDIX was addressed by drawing on the infrastructure adoption score and the socioeconomic score [42]. The 2021 version of digital divide index highlights the "modern version of the knowledge gap" [43], representing the ratio which ranges from 0 to 100, where the higher the number, the higher is the digital divide.

Considered as a sustainability indicator, DDIX shows the potential of the digital inclusion [41,44,45]. These facts led to consider it as a key-factor for the digitainability modelling. Thus, it has been used as an important endogenous factor in the econometric Model 1 and Model 2. The results astonished the authors with the global significance of the equations, which led them to create a third model, by considering DDIX as exogeneous variable.

The raw data for DDIX were extracted from the literature for 13 developed countries from the European Union [43] (Figure 3). The authors used this dataset because it compounds the proportions for 2021 digital divide index calculated by considering the digital gap and digital divide index as a combination that offers more relevance and complexity of this study. Moreover, this method ensures a continuation of a significant previous study that demonstrated that education is an independent variable with high impact on Internet usage. According to data, the country with the highest digital divide index is Finland and the country with the lowest digital divide is Luxembourg.

## Digital divide index in European Union

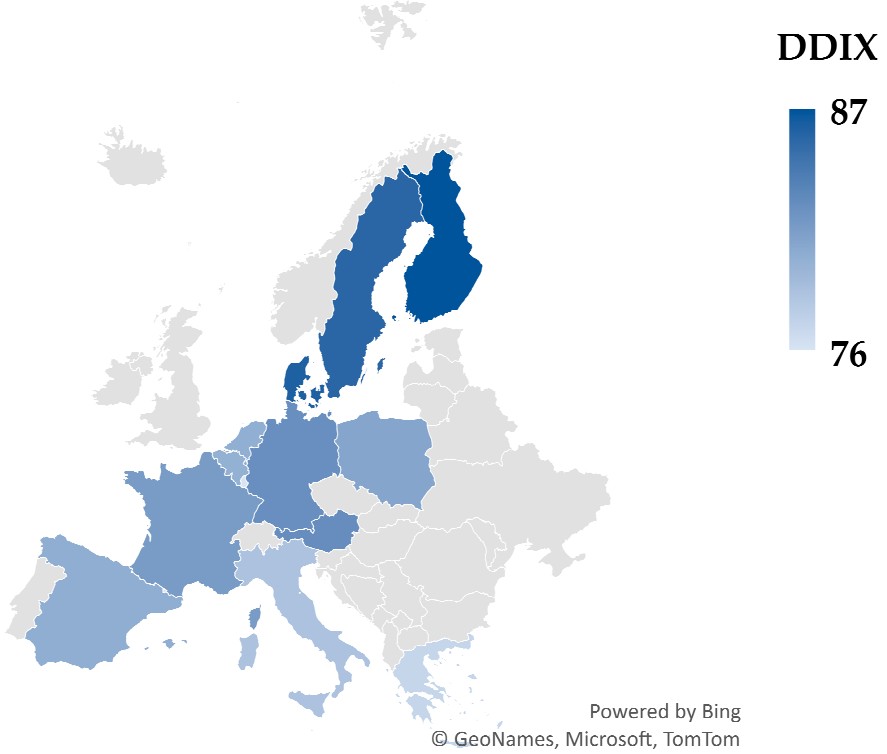

**Figure 3.** Digital Divide Index in EU. Source: authors' conceptualization based on the dataset. Source: authors' conceptualisation.

By integrating the data for all necessary variables to run the regressions, three econometric models were created. Model 1 and model 2 use DDIX as an endogenous variable, explained by several variables, treated as exogenous variables. The third one explains the sustainable development goals index by the digital divide index, used as dependent variable in the previous models. The list of them and the data sources are presented in Table 1.

Thus, based on the knowledge framework, the functional form for the first econometric model is log type and is represented as follows:

$$lnDDIX = \beta_0 + \beta_1 lnWiD + \beta_2 lnRiD + \beta_3 lnDS + \beta_4 lnCCS + \epsilon \tag{1}$$

where $lnDDIX$ represents the natural log of digital divide index, and the same for SDG4, $WiD$, $RiD$, $DS$, $CCS$.

Model 2 conceptualization:

$$lnDDIX = \beta_0 + \beta_1 lnCCS + \beta_2 lnS + \beta_3 lnMhS + \epsilon \tag{2}$$

where $lnDDIX$ represents the natural log of digital divide index, and the same for $CCS$, $S$, $MhS$.

Consulting the functional form of the equations it can be said that the log–log models describe the non-linear relation for these two cases, which are different from the third one, realized with a linear equation.

**Table 1.** The variables of econometric models.

| Model | Variable | Description | Notation | Data Source |
|---|---|---|---|---|
| 1, 3 | Sustainable development goal number 4 (quality education) | Indicator to measure the inclusive and equitable quality education. | SDG4 | Sustainable Development Report 2022 [46] |
| | Women in digital score | Weighted average calculated based on three dimensions: (i) Internet usage, (ii) skills for Internet usage, (iii) specialized skills and employment). | WiD | European Commission, [47] |
| | Rural Digital Index | Score for Internet usage, human capital, and connectivity. | RiD | European Commission, [47] |
| | Digital Skills | % of individuals with above basic overall digital skills. | DS | European Commission, [47] |
| 2 | Communication and collaboration | Above basic skills related to communication and collaboration. | CCS | European Commission, [47] |
| | Security concerns | % of individuals with security concerns when they are buying online. | S | European Commission, [47] |
| | Medium or high computer skills | % of individuals with medium or high computer skills that are running three or more of six computer activities. | MhS | European Commission, [47] |
| 3 | Sustainable Development Goals Index | 2022 SDG Index score | SDGIX | Sustainable Development Report 2022 [46] |

*3.2. Modelling the Sustainable Development Goal Index*

Regarding the third model, the appropriate relation is a linear one, so the equation is presented as follows:

$$SDGIX = \beta_0 + \beta_1 WiD + \beta_2 RiD + \beta_3 DDIX + \epsilon \tag{3}$$

All the models were estimated in EViews software and the results of all three models were crossed to identify a relevant contribution to a conceptual model of digitainability of education (Figure 4).

The results were shaped in the next section and conclusions were brought to light for a better understanding of all concepts linked to the new society within education need to be enhanced, aiming to mitigate the digital divide.

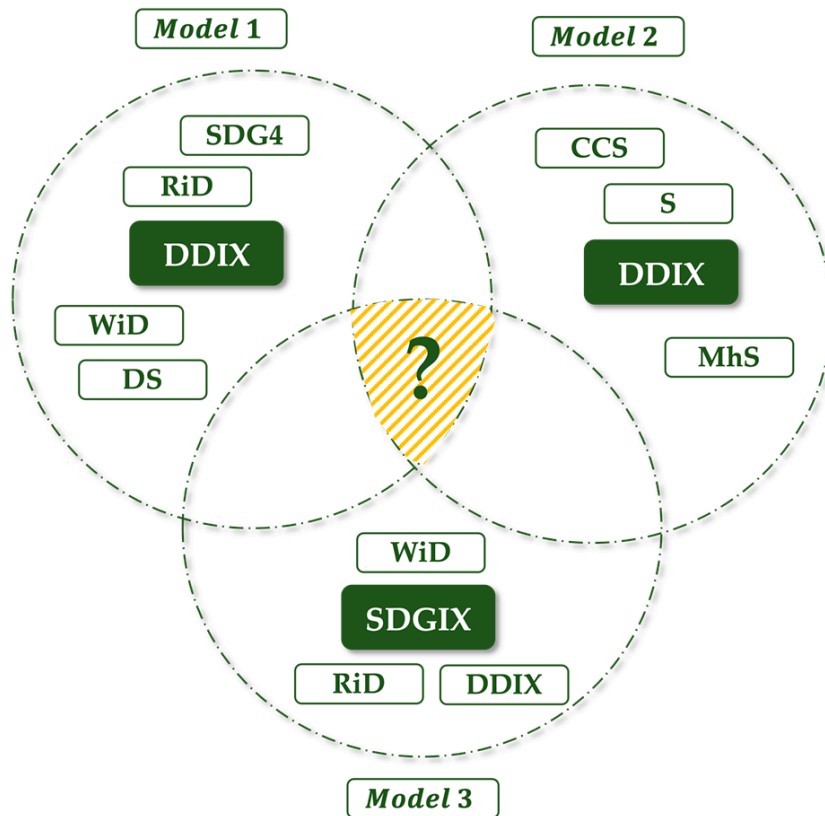

**Figure 4.** Research gap regarding the digitainability of education.

## 4. Results

According to the equation presented in the methodology for the first model, the estimation showed the following results:

$$
\begin{aligned}
\widehat{lnDDIX} = -10.69 &+ 2.27lnSDG4 + 0.69lnWiD + 0.39lnRiD - 0.08lnDS \\
&\quad (2.77) \qquad (0.67) \qquad (0.26) \quad (0.25) \\
&\quad t = 0.81 \qquad 1.02 \qquad 1.47 \quad -0.33 \\
&\quad R^2 = 0.68 \quad \overline{R}^2 = 0.53 \quad N = 13
\end{aligned} \tag{4}
$$

The intercept ($-10.69$) indicates the value of $lnDDIX$ if in a theoretical scenario all the independent variables would be null. The first slope coefficient determines that if SDG4 would modify by 1%, then DDIX would increase with 2.27 points. This fact would represent the opposite of what the authors expected. Trying to test the significance of SDG4 variable, it can be said that it is not statistically significant due to $p - value = 0.43 > 0.05$. In fact, none of the variables was identified as significant in this model. However, an important result is highlighted by the existence of the global significance of the equation (F-statistic = 0.035075 < 0.05) that shows the relevance of the model. Moreover, the coefficient of determination, $R^2 = 0.68$, as indicator that shows the good fitting of the model, shows that the variability of DDIX around its mean is explained by the SDG4, WiD, RiD, and DS in proportion of 68%, which is justifiable for 13 so different countries.

The second model brings to the light a special result. Besides the good value of fitting indicator, $R^2$ and the statistical global significance supported by F-statistic = 0.004069 < 0.05

(significant even for a higher level of confidence, 99% ($\alpha = 1\%$), one of the three explanatory variables shows statistical significance.

$$\widehat{lnDDIX} = -1.28 + 0.31lnCCS + 0.18lnS + 0.78lnMhS$$
$$(0.59) \qquad (0.1) \qquad (0.31)$$
$$t = 0.53 \qquad 1.74 \qquad 2.48$$
$$R^2 = 0.75 \quad \overline{R}^2 = 0.67 \quad N = 13$$

(5)

The medium or high computer skills variable presents a *p*-value equal to 0.0349, which indicates that for $\alpha = 5\%$, it is statistically significant (*p*-value $< \alpha$). Interpreting the significance of slope coefficient, $\beta_3$, it can be said that a 1% change in individuals with medium or high computer skills generates percentage increase in digital divide index with 0.78 points.

The results of the third model are presented below:

$$SDGIX = 71.33 + 0.09WiD - 14.71RiD + 0.24DDIX$$
$$(0.11) \qquad (8.6) \qquad (0.1)$$
$$t = 0.87 \qquad -1.74 \qquad 2.28$$
$$R^2 = 0.56 \quad \overline{R}^2 = 0.41 \quad N = 13$$

(6)

The global significance of the model is ensured by the value of *F-statistic* $= 0.049912 < p$-value $= 0.05$, but in this case, as well, the significance of one variable drew the attention of the authors. It was DDIX which showed that it positively influences the SDGIX variable. For a better understanding the researchers represented graphically the regressions and *p*-values, and it can be observed that SDGIX is explained by DDIX, which in turn is explained by MhS. This means that the digital divide index influences sustainable development goals index score directly, in positive way ($\beta_3 = 0.242685$). Moreover, if the result obtained from the estimation of the Model 2 is added in the model, the researchers can admit that indirectly, the medium or high variable relates to SDGIX through the digital divide index, as it is shown in Figure 5.

The arrows in the diagram illustrating the model have been drawn for the purpose of showing the relations between variables for each model that has been conceptualised. The independent variables were connected in relation to the endogenous variable according to the estimation of the equations. Therefore, after the estimation process the specific numbers of *p*-values and intercepts were added. Consulting the blue colours, it can be stated with ease that only two of them were significant for the model. Considering this result, the authors have gone further and have created another diagram to formulate the third hypothesis by considering that variables from the models 1 and 2, (globally significant) can influence the sustainable development score (Figure 6).

On one hand, the intersection of the models shows the possibility of DDIX explained in the first row by SDG4, RiD, WiD, and DS and in the second row explained by CCS, S, Mhs, to indirectly shape the inclusivity in the new society by enhancing the digitainability of sustainable development goals with education. The illustration of the two transparent arrows at the top of the diagram graphically supports this statement, but on the other hand, the solid-coloured arrows from the bottom show the demonstrated connection between variables, namely, MhS and SDGIX with digitainability of sustainable development goals with education.

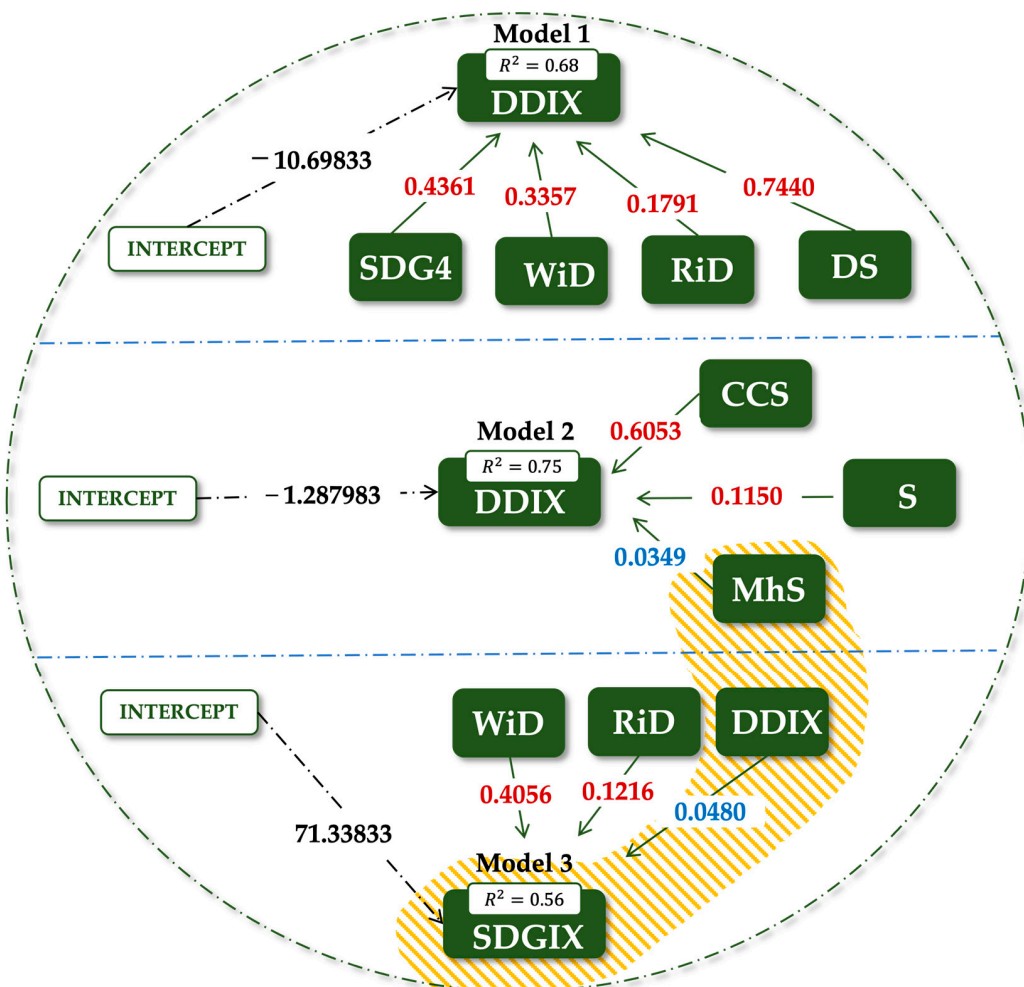

**Figure 5.** Digitainability model of sustainable development goals. Source: authors' conceptualization based on results.

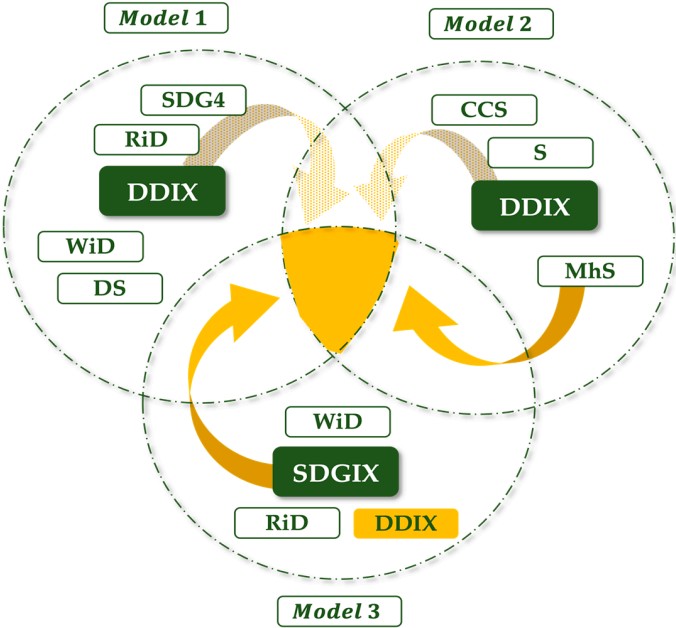

**Figure 6.** The result obtained by crossing the econometric models.

## 5. Discussion

The literature presents many perspectives related to education and sustainable development goals because education has impacted all SDGs except SDG 14 [25,26]. Typing the expression on Google Academic, only in this platform 16,700 results were displayed. In such a context, it is difficult to add something new or to perform a study that cannot be linked with another one. This is the reason why the authors do not want to present this study as a new one, or a paper that brings something unfaced. They want to increase the level of awareness of the truth behind the accelerated rhythm of technological development and the defining characteristic of the Society 5.0. The digital divide index, linked to sustainable development goals into a single big picture, can show the real challenge that the future brings to the world. Joining the sustainability and digitalisation and presenting it as digitainability of a phenomenon can deeply enhance the analysis methods. In this paper, the researchers chose to study the fourth sustainable development goal related to education to present a determining truth about how the index that compounds all the goals can be modelled.

By the instrumentality of research questions formulated, a total of six objectives were addressed. They were formulated to cover vital assets for sustainable development in the context of evolving to a new and inclusive Society [19]. Some of them were achieved and expected to be described with the results obtained. Otherwise, some of them have unveiled details that challenged the researchers to think forward. Intending to model the digital divide index, two different equations were specified. The first one has placed the fourth sustainable development goal in a special position besides women in digital score, rural digital index, and digital skills. Together, all four variables are very important for digital divide and their changes can contribute to achieving the ideal inclusive society by education. The same index, modelled by communication and collaboration skills, security concerns, and medium or high computer skills proves the importance of these skills for future generations to be integrated and included for the next jobs and specialisations. Skills that cannot be missing in society 5.0, they are considered even competitive advantages and target goals that can be achieved through education as well. SDG4 is a critical goal for an inclusive future and its importance on the 2030 Agenda highlights the scientific contribution of this study. Digital transformation, digitisation, and sustainability must be connected to the education process because the technological changes create gaps, not only generational, but also behavioural [35]. Understanding this issue is a first step in creating a technologically inclusive environment and human-centred Industry, as Society 5.0 requires [2,10]. Moreover, digitalisation can contribute to resilience in some critical cases or crisis [3,13,14], which highlights the enabled-sustainable behaviour in helping individuals, no matter the age or development stages in which different EU member states are. Education is so important to ensure the sustainability and digitalisation that it can generate societal changes based on socio-technical transformation behaviours that enlarge the digital divide, one of the most important challenges for the New Society. Considering the result that highlights the indirect relation that medium or high computer skills have with sustainable development goals overall score through the digital index, it is proved once again the fact that educating the people to have developed skills in this field can be a positive influence on digital divide mitigation and incremental inclusiveness.

Arguing with the existence of mentions in specialized literature regarding the individuals with special integration needs in a continuously evolving Society, the authors recall the role played by higher education in adapting to a technological-societal development. However, and above all, the training of teachers and educators remains imperative and their skills above average in computer usage are considered a must-have skill to create the future of Society 5.0 through education.

## 6. Conclusions

The digitainability of sustainable development goals seems to be a real challenge at the first sight. However, through education the progress is just a matter of time. By deploying

relevant policies that can only enhance the inclusivity in the Society 5.0 will diminish the challenges that humanity faces every day. For this purpose, the academics must understand and promote scientific contributions regarding the concept of digitainability and continue to search new angles and perspectives from which to look for a better, inclusive, and sustainable future.

The major contribution of this paper to the literature is represented by the model of digitainability, which is influenced by medium or high computer skills and sustainable development index. This outcome confirms the necessity of optimising the teaching and learning process with high standards for digital skills. This fact conducts the author to theoretical implications that mark the academics, educational institutions, and policy makers to align the solutions found and proven to enhance the rhythm of adoption of a society where technological and human values are inter-connected. The educational public system must offer opportunities for everyone to learn and prepare specialists for high-demanding markets in which they will work, even if special needs must be covered.

The orientation of managerial implications of top leaders from companies must address an adapted mindset to collaborate with the academic environment to discover and understand the truth about the challenges of the new Era. Moreover, their rights and obligations should be bent on continuous professional forming of human resources through reasonable rewards and promotion facilities that cannot conflict the careers of the others. Understanding the needs and capabilities of employees, the managers become responsible for digital divide mitigation in companies by assessing roles that are synchronised with the training level of individual. Over time, they should find suitable resources and identify opportunities for everyone involved in company to fruitful their potential.

A first limitation of the study emerges from the phase of specifying the econometric models. Although the vital role of this type of analysis method in studying economic phenomena is recognised, the choice of the variables is possible to have been made incompletely. This fact can determine the possibility of omitting one or many important variables in the models created, which would represent a limit for research performed, a limit that can be mitigated by the research perspectives. Moreover, referring to the specification model, a limitation of the study might represent the selected functional form of the equations, which conducted the authors to formulate the second limitation.

Third, the sample size used to perform the study is relatively small, due to the actual nature of the 2021 digital divide index indicator which could not allow it to be calculated for other countries. This aspect might generate challenges regarding the homoskedasticity quality of the models in the moment of resizing the sample. Hence, the validity of tests could be more rigorous for larger samples.

Finally, another limitation is represented by secondary data usage instead of direct research and data collection that did not offer more opportunities for analysis to the authors.

Future research can be performed to run the Breusch–Pagan or White test to analyse the variance when sample size is changed. Moreover, the results of these tests would give the opportunity to identify if significant variables would be missing from the econometric models. Another perspective on this study can be addressed in the future if direct marketing research would be running to understand the attitude and opinions of academicians and managers about digital divide and sustainable development through education.

**Author Contributions:** Conceptualization, L.D., E.N., I.B.C. and G.B.; methodology, E.N. and G.B.; literature review, I.B.C. and L.D.; analysis and writing the results, E.N.; discussion and conclusions, E.N., I.B.C., L.D. and G.B.; writing—original draft preparation, L.D., E.N. and I.B.C.; writing—review and editing, L.D., I.B.C., E.N. and G.B.; supervision, G.B.; project administration, G.B.; funding acquisition, G.B. All authors have read and agreed to the published version of the manuscript.

**Funding:** The APC was funded by Transylvania University of Brasov.

**Institutional Review Board Statement:** Not applicable.

**Informed Consent Statement:** Not applicable.

**Data Availability Statement:** Data used for this study are available online at: https://bit.ly/3Y8Vl0l (accessed on 28 January 2023).

**Conflicts of Interest:** The authors declare no conflict of interest.

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
