# Peer review of "Shaping the Inclusivity in the New Society by Enhancing the Digitainability of Sustainable Development Goals with Education"

_sustainability, doi:10.3390/su15043782_

Round 1
Reviewer 1 Report
The paper presents a new perspective on sustainable development goals analysis focusing on the concept of digitainability and seven indicators with a view to education and inclusivity.
Clear presentation of the framework of the study and research methodology. Good presentation and discussion on the current trends in industry, technology and society in relation to education and inclusivity.
More explanations are needed on the presented diagrams, especially the significance of arrows.
Conclusions need to be deepened and more precise. For example, this is an expected result with a view to the digitalisation of industry, and cannot be claimed as unexpected nor significant:
Line 334 "Taking into account the unexpected result that highlights the indirect relation that medium or high computer skills have with sustainable development goals overall score through digital index..."
Also, this conclusion is very general and not directly related to your study:
line 344 "For this purpose, the academics must seek the truth, present it to the world as it is, undeformed and objective."
Additionally, the educational aspects are underrepresented in the discussion and the conclusions.
Minor language issues and punctuation:
line 40 defining “Industry 6.0 [2].
line 57 to understand how sustainable, and digitalisation can create more inclusivity in the Society
Figure 1 - correct the research questions, do you mean "What influenceS the digital divide index?"
line 257 brought in the light - brought to light
line 273 acceptable for such 13 different countries
line 300 recalling the Figure 4 - recalling Figure 4
Alignment in (4) after line 261
The two diagrams in Fig. 6 are overlapping
Reviewer 2 Report
Dear Authors,
I am pleased to review the paper entitled "Shaping the Inclusivity in the New Society by Enhancing the Digitainability of Sustainable Development Goals with Education" and proposed three econometric models at an intersection of education and social inclusivity. Although the idea and topic of this work are interesting, the manuscript must be improved. I serve a few of my suggestions below:
1. The abstract is quite informative, however, the research objective is totally missing. It focuses on the findings, but at the beginning, the study's rationale and OBJECTIVE must be articulated.
2. Theoretical and practical implications of the study can be the right extent to the final part of the abstract as well.
"Using" looks like a tautology, please rephrase.
3. (27-36) The introduction looks well-written, however, in order to make it more engaging to a reader and to give a deeper and wider picture of the current context, I encourage authors to add a second paragraph dedicated to a cross-field introduction of how technology makes fields of society more inclusive. Notably, I would highly recommend observing the qualitative study in open access:
https://www.frontiersin.org/articles/10.3389/fpsyg.2022.805043/full
Glebova, E., Desbordes, M. and Geczi, G., 2022. Mass Diffusion of Modern Digital Technologies as the Main Driver of Change in Sports-Spectating Audiences. Frontiers in Psychology, 13.
I also recommend going cross-field and bringing short examples:
Pohawpatchoko, C., Colwell, C., Powell, J. and Lassos, J., 2017. Developing a native digital voice: Technology and inclusivity in museums. Museum Anthropology, 40(1), pp.52-64.
4. figure 1 could be improved in quality, please
5. (85-87) Following my recommendation about the abstract, I would invite the authors to state the study objective at the end of the introduction, before the lit review. This is crucial.
6. (188) Please cite SDG 4 in the text
7. (323) a few citations would strengthen the opening statement. Who are "many"? Please indicate them namely via citations...
8. The discussion should be more coherent with the literature review. Please build more logical bridges between the theoretical part and the discussion
9. Please extend limitations, there is definitely more to say about study limits, refer to the methodological literature as well.
Round 2
Reviewer 2 Report
Dear Authors,
Thank you for the effective revisions.